



# $^1$H enhanced $^{103}$Rh NMR spectroscopy and relaxometry of Rh(acac)$_3$ in solution

Harry Harbor-Collins[1], Mohamed Sabba[1], Markus Leutzsch[2], and Malcolm H. Levitt[1]

[1]School of Chemistry, University of Southampton, SO17 1BJ, UK
[2]Max-Planck-Institut für Kohlenforschung, Kaiser-Wilhelm-Platz 1, Mülheim an der Ruhr, 45470, Germany

**Correspondence:** Malcolm H. Levitt (mhl@soton.ac.uk)

**Abstract.** Recently-developed polarisation-transfer techniques are applied to the $^{103}$Rh Nuclear Magnetic Resonance (NMR) of the Rh(acac)$_3$ coordination complex in solution. Four-bond $^1$H-$^{103}$Rh J-couplings of around 0.39 Hz are exploited to enhance the $^{103}$Rh NMR signal and to estimate the $^{103}$Rh $T_1$ and $T_2$ relaxation times as a function of field and temperature. The $^{103}$Rh longitudinal $T_1$ relaxation in Rh(acac)$_3$ is shown to be dominated by the spin-rotation mechanism, with an additional field-dependent contribution from the $^{103}$Rh chemical shift anisotropy.

## 1 Introduction

Although rhodium is one of the few chemical elements with a 100% abundant spin-1/2 isotope, the routine NMR of $^{103}$Rh has been inhibited by its very small gyromagnetic ratio, which is negative and $\sim 31.59$ times less than that of $^1$H (Mann (1991)).

While indirectly-detected $^{103}$Rh NMR has had an appreciable history (Crocker et al. (1979); Dykstra et al. (1981); Brevard and Schimpf (1982); Heaton et al. (1983); Brevard et al. (1981); Herberhold et al. (1999); Ernsting et al. (2004); Carlton (2008)), advances in instrumentation and methodology have allowed rapid observation of $^{103}$Rh NMR parameters on standard commercial NMR spectrometers, leading to a recent renaissance of the field (Chan et al. (2020); Bajo et al. (2020); Wiedemair et al. (2021); Widemann et al. (2021); Rösler et al. (2021); Sheng Loong Tan et al. (2021); Caló et al. (2021); Samultsev et al. (2022); Harbor-Collins et al. (2023); Holmes et al. (2023); Lutz et al. (2023); Harbor-Collins et al. (2024)).

The rhodium (III) acetylacetonate complex (Rh(acac)$_3$, see figure 1) currently serves as the IUPAC $^{103}$Rh NMR chemical shift reference (Carlton (2008)). To the wider scientific community, Rh(acac)$_3$ is better known for its role in the production of thin rhodium films and nanocrystals for use in catalysis (Zhang et al. (2007); Aaltonen et al. (2005); Choi et al. (2016)).

The early studies of nuclear spin relaxation in Rh(acac)$_3$ were greatly limited by the poor $^{103}$Rh signal strength, and provided somewhat conflicting conclusions for the $^{103}$Rh relaxation mechanisms (Grüninger et al. (1980); Benn et al. (1985); Maurer et al. (1982)). Recently, a two-bond $^{13}$C-$^{103}$Rh coupling of 1.1 Hz was observed in Rh(acac)$_3$, and was exploited for triple-resonance experiments (Caló et al. (2021)).

We now report the observation of a four-bond $^1$H-$^{103}$Rh J-coupling of $|^4\mathrm{J}_{\mathrm{HRh}}| \simeq 0.39\,\mathrm{Hz}$ between the central $^{103}$Rh nucleus and each of the three methine $^1$H nuclei in Rh(acac)$_3$ (see figure 1). These small couplings are exploited for the $^1$H-enhanced $^{103}$Rh NMR spectroscopy of the Rh(acac)$_3$ complex. $^{103}$Rh spin-lattice $T_1$ and spin-spin $T_2$ relaxation time constants are





measured over a range of magnetic fields and temperatures. The $^{103}$Rh $T_1$ relaxation is found to be dominated by spin rotation, with an additional contribution from the CSA, which is significant at high fields.

## 2  Experimental

Experiments were performed on a saturated ($\sim 140$ mM) solution of rhodium (III) acetylacetonate (Rh(acac)$_3$) dissolved in 350 $\mu$L CDCl$_3$. Rh(acac)$_3$ was purchased from Sigma-Aldrich and used as received. Rh(acac)$_3$ is a bright yellow-orange

powder, which dissolved in CDCl$_3$ to form a solution with a deep golden colour.

The radiofrequency channels were additionally isolated by installing a bandpass (K&L Microwave) and low-pass (Chemagnetics 30 MHz) filter at the preamplifier outputs of the $^1$H and $^{103}$Rh channels respectively. Pulse powers on the $^1$H and $^{103}$Rh channels were calibrated to give a matched nutation frequency of $2\pi \times 4000$ Hz corresponding to a 90° pulse length of 62.5 $\mu$s. Field cycling experiments were performed using a motorised fast shuttling system (Zhukov et al. (2018); Harbor-Collins

et al. (2023, 2024)). The shuttling time was kept constant at 2 seconds, in both directions.

**Figure 1.** The molecular structure of rhodium (III) acetylacetonate, Rh(acac)$_3$, which has point group symmetry D$_3$. This work exploits the long-range $^4J_{\text{HRh}}$ scalar couplings for polarisation transfer between the $^{103}$Rh and methine $^1$H spins.





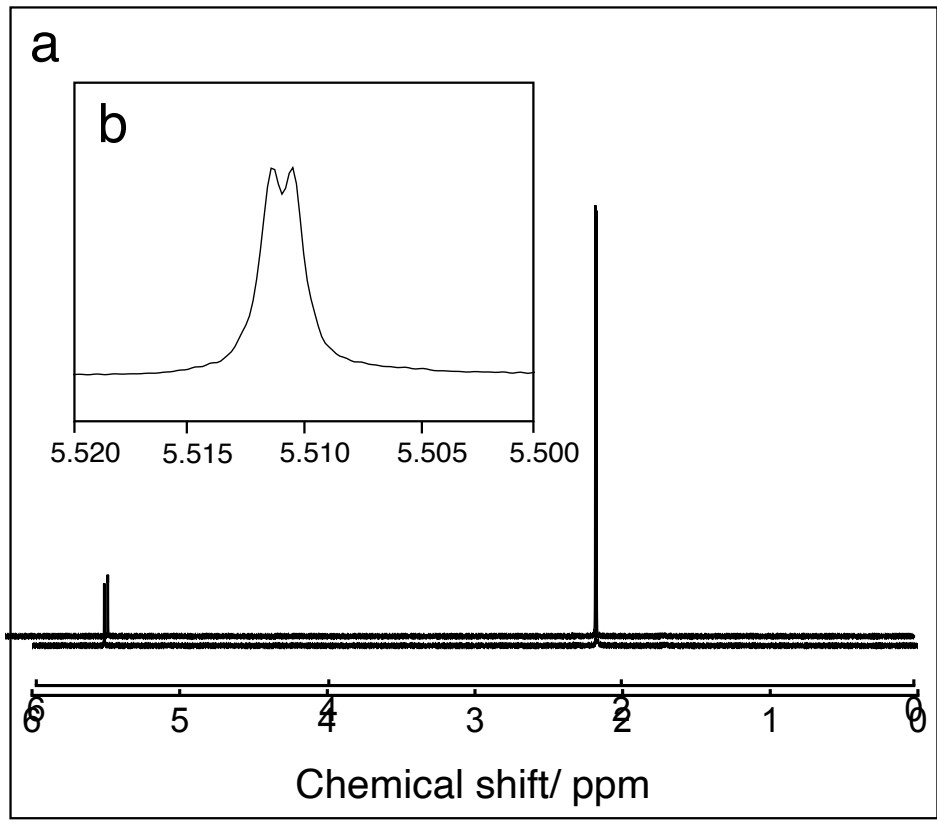

**Figure 2.** (a) $^1$H spectrum of a $\sim$140 mM solution of $\mathrm{Rh(acac)_3}$ in $\mathrm{CDCl_3}$, acquired at 9.4 T and at 298K, acquired in a single transient. Lorentzian line broadening (1 Hz) was applied (b) Expanded view of the methine $^1$H resonance. Negative Lorentzian line broadening (-0.2 Hz) was applied to enhance the resolution.

## 3 Results

### 3.1 $^1$H spectrum

The $^1$H spectrum for $\mathrm{Rh(acac)_3}$ in $\mathrm{CDCl_3}$ shown in Figure 2 (a) features two resonances: a singlet at 2.170 ppm corresponding to the six methyl protons on each acac ligand and a broad, weak doublet centred at 5.511 ppm corresponding to the acac methine protons. An expanded region showing just the methine resonance is shown in Figure 2 (b). The four-bond $^{103}$Rh-$^1$H spin-spin coupling is estimated to be $|^4 J_{\mathrm{HRh}}| = 0.39 \pm 0.01$ Hz.



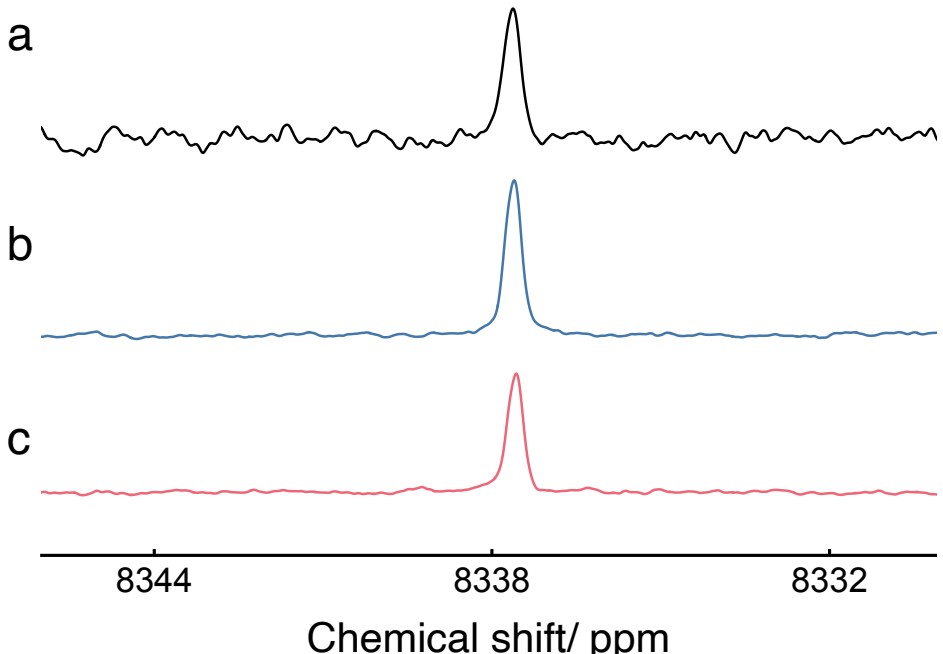

**Figure 3.** $^1$H-decoupled $^{103}$Rh NMR spectra of a $\sim$140 mM solution of $\mathrm{Rh(acac)_3}$ in CDCl$_3$, in a field of 9.4 T and a temperature of 295 K. Lorentzian line broadening (1 Hz) was applied to all spectra. $^{103}$Rh chemical shifts are referenced to the absolute frequency ($\Xi\left(^{103}\mathrm{Rh}\right)$ = 3.16%). In all spectra, $^1$H decoupling was achieved using continuous wave decoupling with 0.05 W of power, corresponding to a nutation frequency of 1 kHz. (a) $^{103}$Rh{$^1$H} spectrum, acquired using 300 transients, each using a single $^{103}$Rh 90° pulse. The waiting interval between transients was 150 seconds. The total experimental duration was $\sim$ 12 hours. (b) $^{103}$Rh{$^1$H} spectrum, acquired using 16 transients and the pulse sequence shown in figure 4, with $n = 11$ repetitions of the DualPol sequence. The waiting interval between transients was 18 seconds. The total experimental duration was 5 minutes. (c) $^{103}$Rh{$^1$H} spectrum, acquired using 16 transients and an optimised refocused-INEPT sequence. The waiting interval between transients was 18 seconds. The total experimental duration was 5 minutes.

## 3.2 $^{103}$Rh spectra

### 3.2.1 Direct $^{103}$Rh excitation

The $^1$H-decoupled $^{103}$Rh spectrum of the $\mathrm{Rh(acac)_3}$ solution, acquired with single-pulse excitation of $^{103}$Rh transverse mag-
netisation, is shown in figure 3(a), and displays a single peak with the $^{103}$Rh chemical shift of 8337.6 ppm. The signal-to-noise ratio is quite poor, even after 12 hours of data acquisition.

### 3.2.2 $^1$H-$^{103}$Rh polarization transfer by DualPol

Polarization transfer from the $^1$H nuclei to the $^{103}$Rh nuclei was performed using the previously-described DualPol pulse sequence incorporating acoustic ringing suppression (Harbor-Collins et al. (2023)), as shown in Figure 4.





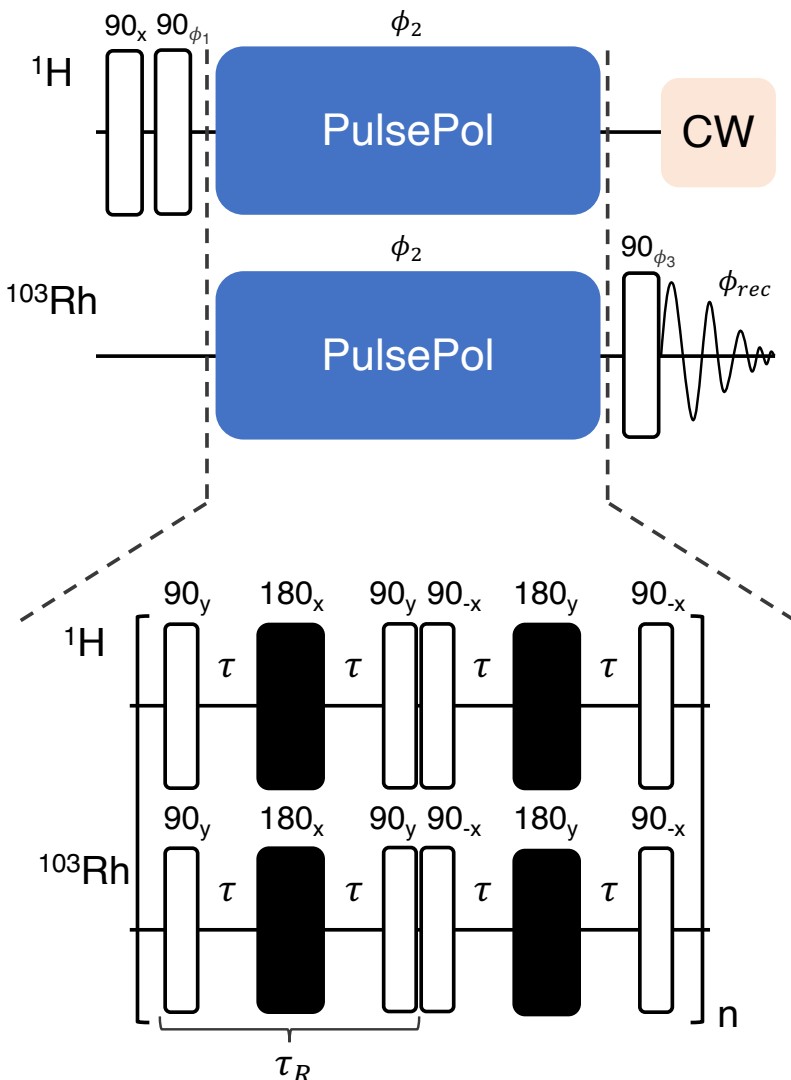

**Figure 4.** Pulse sequence for the acquisition of $^1$H enhanced $^{103}$Rh spectra where a expanded view of the DualPol pulse sequence module is shown at the bottom. The black rectangles indicate symmetrized BB1 composite 180°-pulses (Wimperis (1994); Cummins et al. (2003)) and $\tau$ indicates an interpulse delay. Phase cycles are given by: $\phi_1 = [-x, x, -x, x]$, $\phi_2 = [x, x, -x, -x]$, $\phi_3 = [x, x, x, x, y, y, y, y, -x, -x, -x, -x, -y, -y, -y, -y]$ and the receiver $\phi_{\text{rec}} = [x, -x, x, -x, y, -y, y, -y, -x, x, -x, x, -y, y, -y, y]$.

The DualPol sequence consists of two synchronised PulsePol sequences (Schwartz et al. (2018)), applied simultaneously on two radio-frequency channels.





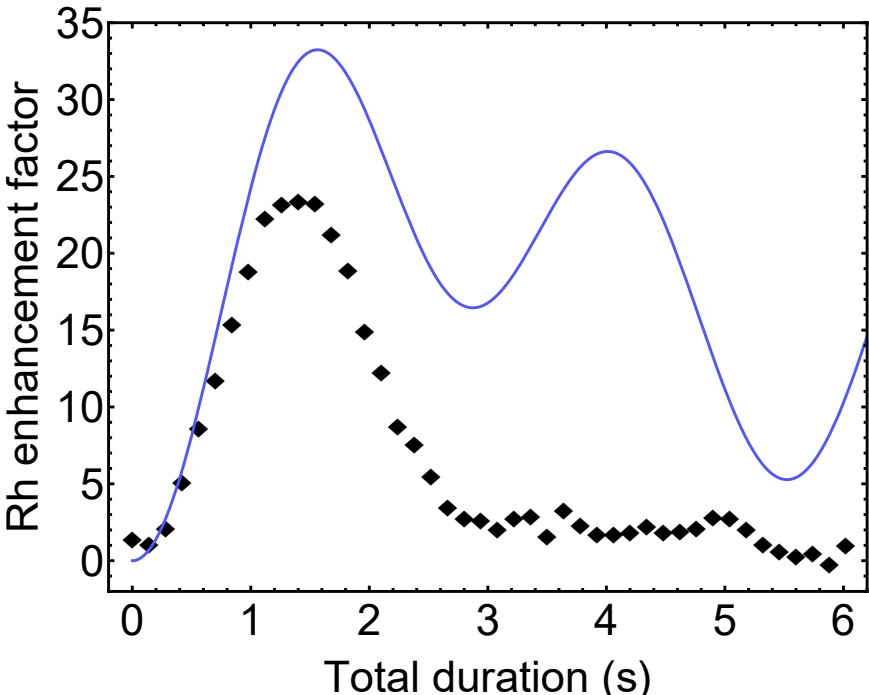

**Figure 5.** $^{103}$Rh signal enhancement factor for Rh(acac)$_3$ as a function of DualPol sequence duration $T$, normalised against thermal-equilibrium $^{103}$Rh polarisation. Black diamonds: experimental data points; solid blue line: the theoretical enhancement factor $|\epsilon_{\mathrm{DualPol}}(T)|$ for an I$_3$S spin system in the absence of relaxation, as given by equation (2) for $|J_{IS}| = 0.39$ Hz.

The PulsePol sequence was originally developed in the context of electron-nucleus polarization transfer (Schwartz et al. (2018)). As discussed in Sabba et al. (2022), PulsePol may be interpreted as a "riffled" implementation of a R-sequence, using the nomenclature of symmetry-based recoupling in solid-state NMR (Carravetta et al. (2000)). In the case of PulsePol, the R-element is a composite $90_y 180_x 90_y$ pulse, with "windows" inserted between the pulses. Furthermore, in the current implementation, the central $180_x$ pulse of each R-element is itself substituted by a BB1 composite pulse (Wimperis (1994)). That substitution was previously shown to increase the robustness of the pulse sequence with respect to deviations in the radiofrequency amplitudes, and resonance offsets (Sabba et al. (2022)). The total R-element duration, including all pulses and "windows", is denoted here $\tau_R$ (see figure 4).

For the experiments described here, the DualPol sequences used an R-element duration equal to $\tau_R = 70$ ms, with pulse durations given by $\tau_{90} = 62.5$ $\mu$s for the $90°$ pulses and $\tau_{\mathrm{BB1}} = 10 \times \tau_{90} = 625$ $\mu$s for the BB1 composite $180°$ pulses.

The $^{103}$Rh and methine $^1$H nuclei of Rh(acac)$_3$ form a I$_3$S spin system, where the $^{103}$Rh nucleus is the S-spin and the magnetically equivalent $^1$H nuclei are the I-spins.



The DualPol spin dynamics are identical to those for Hartmann-Hahn J-cross-polarization (Chingas et al. (1981)). The DualPol average Hamiltonian has the form

$$\overline{H}^{(1)} = \kappa \times 2\pi J_{IS} \left( I_x S_x + I_y S_y \right) \tag{1}$$

where the scaling factor is given by $\kappa = 1/2$ in the limit of short, ideal, radiofrequency pulses. In the absence of relaxation and pulse imperfections, a DualPol sequence with scaling factor $\kappa = 1/2$, applied to a $I_3S$ spin system should give rise to the following enhancement of the S-spin magnetization, relative to its thermal equilibrium value:

$$\epsilon_{\text{DualPol}}(T) = \frac{\gamma_I}{4\gamma_S} \left\{ 2\sin^2\left(\tfrac{1}{2}J_{IS}T\right) + \sin^2\left(\pi J_{IS}T\right) + 2\sin^2\left(\tfrac{1}{2}\sqrt{3}\pi J_{IS}T\right) \right\} \tag{2}$$

Here $T$ is the overall duration of the DualPol sequence. Since the magnetogyric ratios of $^{103}$Rh and $^1$H have opposite signs, the function $\epsilon_{\text{DualPol}}(T)$ is negative for all values of $T$. The blue curve in figure 5 shows a plot of $|\epsilon_{\text{DualPol}}|$ against $T$, for a J-coupling of $|J_{IS}| \simeq 0.39\,\text{Hz}$. The maximum value of $|\epsilon_{\text{DualPol}}|$ is given in the absence of relaxation by

$$|\epsilon_{\text{DualPol}}^{\text{max}}| = |17\gamma_I/16\gamma_S| \simeq 33.56 \tag{3}$$

for the case of I=$^1$H and S=$^{103}$Rh. Since equation (2) is quasi-periodic (Chingas et al. (1981)), the value of $T$ which maximises $|\epsilon_{\text{DualPol}}|$ is indeterminate. The first maximum may be found numerically and occurs at the duration $T_{1\text{st max}} \simeq 0.6098 J_{IS}^{-1}$, at which point the theoretical enhancement is given by $|\epsilon_{\text{DualPol}}| \simeq 1.052|\gamma_I/\gamma_S| \simeq 33.23$ for the case of $I=^1$H and $S=^{103}$Rh. Hence, for the estimated $^1$H-$^{103}$Rh J-coupling of $|J_{IS}| \simeq 0.39\,\text{Hz}$, and assuming $\kappa = 1/2$, the $^{103}$Rh signal enhancement is expected to reach its first maximum at a DualPol duration of $T_{1\text{st max}} \simeq 1.563\,\text{s}$, in the absence of relaxation.

In the experiments described here, the optimum duration of the DualPol sequence was found for a repetition number of $n = 11$. For an R-element duration of $\tau_R = 70\,\text{ms}$, this corresponds to a total DualPol sequence duration of $T = 1.54\,\text{s}$, which is in good agreement with the theoretical value.

The DualPol-enhanced $^{103}$Rh spectrum is shown in figure 3(b), and displays an experimental signal enhancement of $\sim 23$ over the directly-excited $^{103}$Rh spectrum in figure 3(a).

Figure 5 shows the experimental $^{103}$Rh signal enhancement factor as a function of the DualPol sequence duration $T$. Although the maximum of the experimental enhancement occurs at a similar position to the maximum of the theoretical curve, there is clearly a strong damping of the enhancement with respect to the duration $T$, leading to loss of intensity at the theoretical maximum. This damping may be associated with transverse relaxation of the $^1$H and $^{103}$Rh transverse magnetization during the polarization transfer process.

### 3.2.3    $^1$H-$^{103}$Rh polarization transfer by Refocused INEPT

Polarization transfer from $^1$H to $^{103}$Rh may also be conducted by the standard refocused-INEPT pulse sequence Burum and Ernst (1980). In this case, the theoretical enhancement of the S-spin magnetization, due to transfer from the I-spins, is given for the $I_3S$ case, in the absence of relaxation and other imperfections, by Burum and Ernst (1980):

$$\epsilon_{\text{RI}}(\tau_1, \tau_2) = \frac{3\gamma_I}{4\gamma_S} \sin(\pi J_{IS}\tau_1)\left(\sin(\pi J_{IS}\tau_2) + \sin(3\pi J_{IS}\tau_2)\right) \tag{4}$$





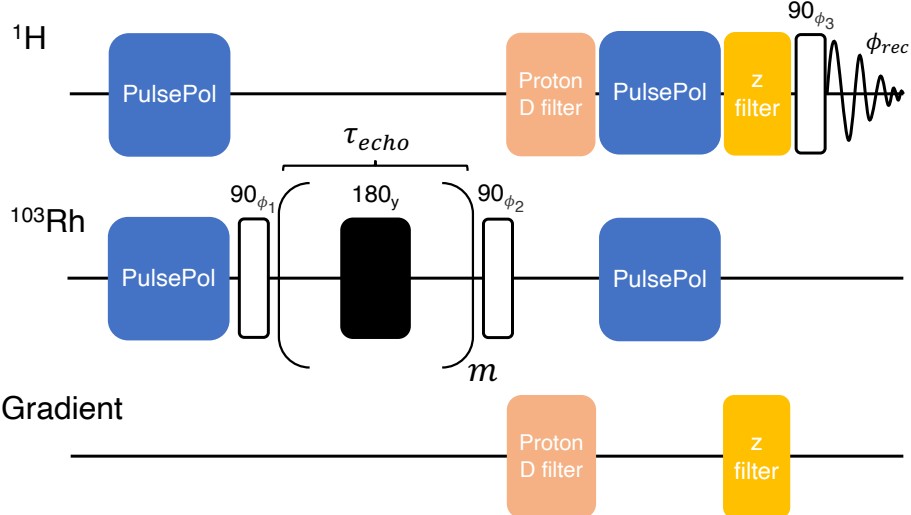

**Figure 6.** Sequence used for the indirect measurement of rhodium $T_2$ through $^1$H detection. The phase cycles are given by $\phi_1 = [x, x, -x, -x]$, $\phi_2 = [-x, x, -x, x]$, $\phi_3 = [x, x, x, x, y, y, y, y, -x, -x, -x, -x, -y, -y, -y, -y]$ and the receiver $\phi_{\mathrm{rec}} = [x, -x, -x, x, y, -y, -y, y, -x, x, x, -x, -y, y, y, -y]$. The echo interval $\tau_{\mathrm{echo}}$ was 45 ms. The black rectangle indicates a symmetrized BB1 composite 180°-pulse (Wimperis (1994); Cummins et al. (2003)). An MLEV-64 supercycle was applied to the phases of the 180° pulses (Levitt et al. (1982a, b); Freeman et al. (1982); Gullion et al. (1990); Gullion (1993)). The $^1$H "D-filter" and "z-filter" modules are described in figures 3 and 4 of reference Harbor-Collins et al. (2023) respectively.

where $\tau_1$ and $\tau_2$ refer to the total echo durations including two inter-pulse intervals, as shown in fig.1 of Burum and Ernst (1980). The maximum of this function is found at $\tau_1 = (2J_{IS})^{-1}$ and $\tau_2 = \arcsin(3^{-1/2})/(\pi J_{IS})$, giving an enhancement of $|\epsilon_{\mathrm{RI}}| = |2\gamma_I/\sqrt{3}\gamma_S|$ (Doddrell et al. (1981, 1982); Pegg et al. (1981, 1982)). The maximum theoretical enhancement by refocused-INEPT is therefore

$$|\epsilon_{\mathrm{RI}}^{\max}| = |2\gamma_I/\sqrt{3}\gamma_S| \simeq 1.155 |\gamma_I/\gamma_S| \simeq 36.48 \qquad (5)$$

for the case of $I=^1$H and $S=^{103}$Rh. Hence, in the absence of relaxation, refocused-INEPT can give a slightly greater enhancement than DualPol in a $I_3S$ system. However, the maximum enhancements by both DualPol and INEPT are less than the theoretical bound on the enhancement of S-spin magnetization by polarization transfer from the I-spins in a permutation-symmetric $I_3S$ spin system, which is equal to $|3\gamma_I/2\gamma_S|$ (Nielsen et al. (1995)).

The theoretical advantage of INEPT over DualPol is not realized in practice, for the case of $\mathrm{Rh(acac)}_3$. The maximum

enhancement by refocused INEPT was realised for durations of $\tau_1 = 920\,\mathrm{ms}$ and $\tau_2 = 500\,\mathrm{ms}$, which yielded a $^{103}$Rh enhancement factor of $\sim 17$ over thermal polarization, i.e. less than the maximum DualPol enhancement, which was $\sim 23$. The experimentally optimised interval $\tau_1$ is significantly shorter than the optimum theoretical value in the absence of relaxation, which is $\tau_1^{\mathrm{theor}} = 1.28\,\mathrm{s}$, assuming a $^{103}$Rh-$^1$H spin-spin coupling of $J_{\mathrm{HRh}} = 0.39$ Hz. The optimum value of $\tau_2$, on the other hand, is very similar to the theoretical value, which is $\tau_2^{\mathrm{theor}} = 503\,\mathrm{ms}$.





The $^{1}$H-enhanced $^{103}$Rh spectrum, produced by an optimised refocused-INEPT sequence, is shown in Figure 3(c). It shows a significantly lower enhancement than the DualPol result of Figure 3(b), despite the fact that the theoretical enhancement by Refocused-INEPT is higher than that of DualPol in the absence of relaxation (see equations 3 and 5). The loss of amplitude relative to the theoretical values may be attributed to transverse $^{1}$H relaxation during the polarization transfer process. It is known that Hartmann-Hahn-style cross-polarisation sequences such as DualPol can outperform INEPT in the presence of

transverse relaxation (Levitt (1991)).

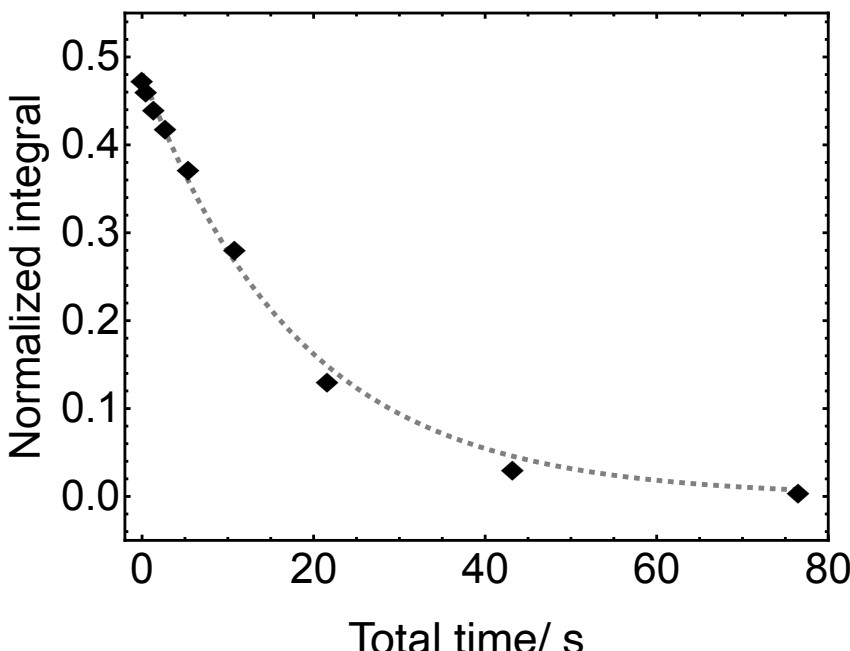

**Figure 7.** Decay curve for the $^{103}$Rh transverse magnetization of Rh(acac)$_3$ in solution at a field of 9.4 T, measured by the indirectly-detected multiple spin-echo scheme in figure 6. The experimental duration was 15 minutes.

## 4    $^{1}$H-detected $^{103}$Rh $T_2$

The $^{103}$Rh $T_2$ relaxation time constant for Rh(acac)$_3$ in CDCl$_3$ was measured via the methine $^{1}$H signals using a variant of a previously described indirectly-detected $T_2$ DualPol pulse sequence (Harbor-Collins et al. (2023)), which is shown in Figure 6. The pulse sequence starts with a DualPol sequence of duration $T = 1.54$ s to transfer thermal-equilibrium longitudinal $^{1}$H

magnetization to $^{103}$Rh. The $^{103}$Rh longitudinal magnetization is converted into $^{103}$Rh transverse magnetization by a 90° pulse on the $^{103}$Rh channel. The $^{103}$Rh transverse magnetization evolves under a Carr-Purcell Meiboom-Gill (CPMG) train




of $m$ spin echoes, each with an echo duration $\tau_{\text{echo}} = 45\,\text{ms}$. The Carr-Purcell sequence suppresses the confounding effects of translational diffusion and mixing with antiphase spin operators (Peng et al. (1991)). The $^{103}\text{Rh}$ transverse magnetization is converted to $^{103}\text{Rh}$ longitudinal magnetization by a second $90°$ $^{103}\text{Rh}$ pulse. A $^1\text{H}$ "D-filter" module is applied to destroy

any residual $^1\text{H}$ magnetisation, before another DualPol sequence of duration $T = 1.54$ s transfers the $^{103}\text{Rh}$ longitudinal magnetization to $^1\text{H}$ longitudinal magnetization. A $^1\text{H}$ "z-filter" module is applied to destroy any other $^1\text{H}$ magnetization components, followed by a $90°$ $^1\text{H}$ pulse which excites $^1\text{H}$ transverse magnetization whose precession induces a $^1\text{H}$ NMR signal which is detected in the following interval. The $^1\text{H}$ "D-filter" and "z-filter" modules are described in figures 3 and 4 of reference Harbor-Collins et al. (2023), respectively.

Repetition of the experiment with increasing values of $m$ leads to the $^{103}\text{Rh}$ $T_2$ decay curve shown in Figure 7. This fits well to a single exponential decay with the time constant $T_2(^{103}\text{Rh}) = 18.36 \pm 0.92$ s.

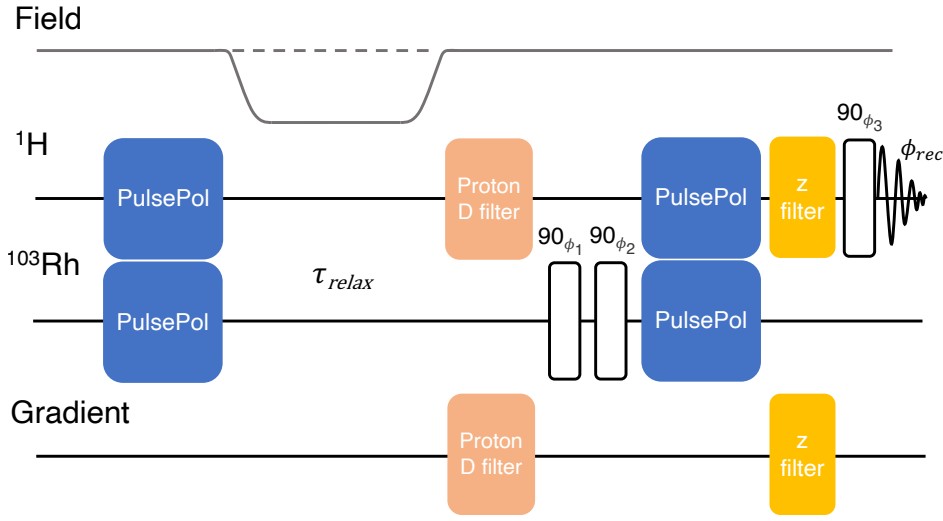

**Figure 8.** Sequence used for the indirect measurement of the rhodium $T_1$ through the $^1\text{H}$ NMR signal. Phase cycles are given by $\phi_1 = [x, x, -x, -x]$, $\phi_2 = [-x, x, -x, x]$, $\phi_3 = [x, x, x, x, y, y, y, y, -x, -x, -x, -x, -y, -y, -y, -y]$ and the receiver $\phi_{rec} = [x, -x, -x, x, y, -y, -y, y, -x, x, x, -x, -y, y, y, -y]$. The optional shuttling of the sample to low field, and back again, during the interval $\tau_{\text{relax}}$, is indicated. The $^1\text{H}$ "D-filter" and "z-filter" modules are described in figures 3 and 4 of reference Harbor-Collins et al. (2023) respectively.

## 5   $^1\text{H}$-detected $^{103}\text{Rh}$ $T_1$

The $^{103}\text{Rh}$ $T_1$ relaxation time constant for $\text{Rh(acac)}_3$ in $\text{CDCl}_3$ was measured indirectly using the methine $^1\text{H}$ signals, by means of the pulse sequence shown in Figure 8 (Harbor-Collins et al. (2023)). The pulse sequence starts with a DualPol

sequence of duration $T = 1.54$ s to transfer thermal-equilibrium longitudinal $^1\text{H}$ magnetization to $^{103}\text{Rh}$. For variable-field experiments, the sample is shuttled out of the high-field magnet into a low-field region. The nuclear magnetization is allowed to





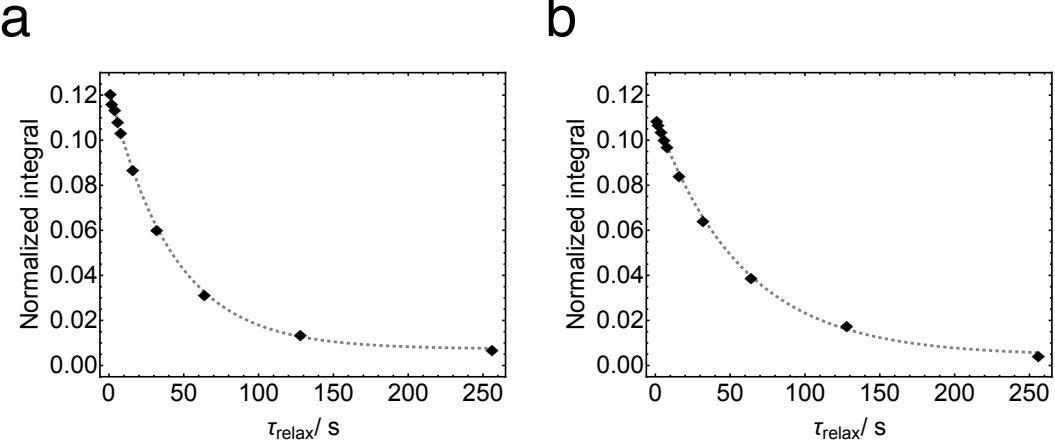

**Figure 9.** Trajectories of longitudinal $^{103}$Rh magnetization for $\mathrm{Rh(acac)_3}$ in solution at a temperature of 295 K, measured indirectly through the methine $^1$H signals, using the pulse sequence in figure 8. (a) Filled symbols: $^1$H signal amplitudes at a magnetic field of 9.4 T. The data was acquired in ∼2 hours. The integrals are normalised against the $^1$H spectrum obtained by a single $^1$H 90° pulse applied to a system in thermal equilibrium at 9.4 T and at 295 K. Dotted line: Fitted exponential decay with time constant $T_1(^{103}\mathrm{Rh}) = 41.8 \pm 0.9$ s. (b) As in (a), but shuttling the sample to a field of 10 mT during the relaxation delay $\tau_{\mathrm{relax}}$. Dotted line: Fitted exponential decay with time constant $T_1(^{103}\mathrm{Rh}) = 57.8 \pm 1.7$ s.

relax for an interval $\tau_{\mathrm{relax}}$. If necessary, the sample is shuttled back into high field, and residual $^1$H magnetization destroyed by a $^1$H "D-filter" module. A pair of phase cycled 90° $^{103}$Rh pulses are applied to select for $^{103}$Rh z-magnetisation before a second DualPol sequence of duration $T = 1.54$ s transfers the partially-relaxed longitudinal $^{103}$Rh magnetization to $^1$H magnetization.

A $^1$H "z-filter" module is applied to destroy any other $^1$H magnetization components, followed by a 90° $^1$H pulse which excites $^1$H transverse magnetization whose precession induces a $^1$H NMR signal which is detected in the following interval. The $^1$H "D-filter" and "z-filter" modules are described in figures 3 and 4 of reference Harbor-Collins et al. (2023), respectively. During $\tau_{\mathrm{relax}}$, $^1$H enhanced $^{103}$Rh magnetisation decays toward thermal $^{103}$Rh magnetisation, which is very small; hence, at large values of $\tau_{\mathrm{relax}}$, resulting $^{103}$Rh derived $^1$H signals are very weak and close to zero, even for measurements performed at

higher magnetic field strengths.

The trajectory of indirectly-detected $^{103}$Rh z-magnetization in a field of 9.4 T and temperature 295 K is shown in Figure 9(a). The trajectory fits well to a single-exponential decay with time constant $T_1(^{103}\mathrm{Rh}) = 41.8 \pm 0.9$ s. A trajectory in the low magnetic field of 10 mT and temperature 295 K is shown in Figure 9(b). This was produced by shuttling the sample to low magnetic field, and back again, during the interval $\tau_{\mathrm{relax}}$. The relaxation process is somewhat slower in low magnetic field, with

a time constant of $T_1(^{103}\mathrm{Rh}) = 57.8 \pm 1.7$ s.





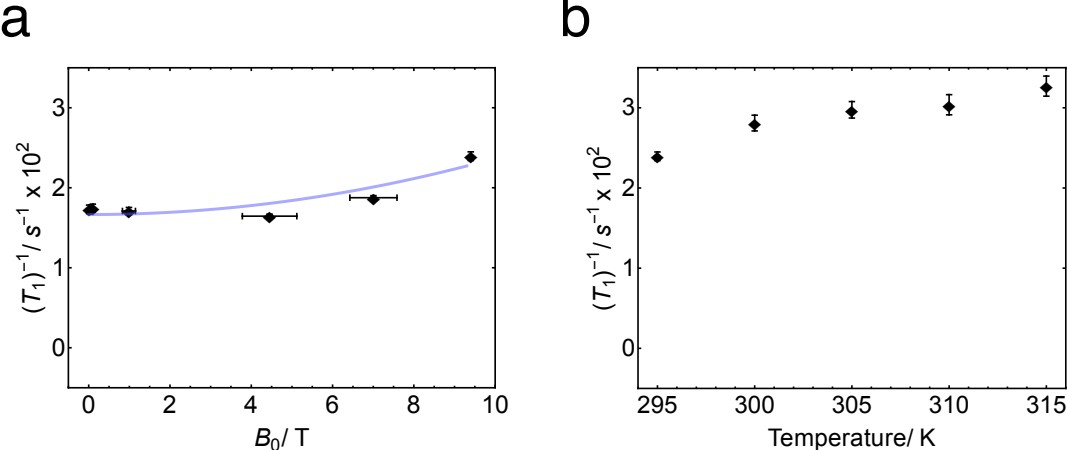

**Figure 10.** (a) $^{103}$Rh relaxation rate constant $T_1^{-1}$ for $\mathrm{Rh(acac)_3}$ in solution, as a function of magnetic field strength at a temperature of 295 K. The blue line shows the best-fit quadratic function $T_1^{-1}(B) = T_1^{-1}(0) + aB^2$, where $T_1^{-1}(0) = (167 \pm 7) \times 10^{-4}\,\mathrm{s}^{-1}$ and $a = (7 \pm 2) \times 10^{-5}\,\mathrm{s}^{-1}\,\mathrm{T}^{-2}$. (b) $^{103}$Rh relaxation rate constant $T_1^{-1}$ for $\mathrm{Rh(acac)_3}$ in solution, as a function of temperature at a magnetic field strength of 9.4 T.

The observed field-dependence of the $^{103}$Rh relaxation rate constant $T_1^{-1}$ is shown in Figure 10(a). The magnetic field-dependence of $T_1^{-1}$ is quite weak in this range of fields. The relaxation rate constant increases slightly with increasing magnetic field at the high-field end, suggestive of a weak relaxation contribution from the $^{103}$Rh chemical shift anisotropy. The blue curve in Figure 10(a) shows the best-fit quadratic function $T_1^{-1}(B) = T_1^{-1}(0) + aB^2$, where $T_1^{-1}(0) = (167 \pm 7) \times 10^{-4}\,\mathrm{s}^{-1}$
and $a = (7 \pm 2) \times 10^{-5}\,\mathrm{s}^{-1}\,\mathrm{T}^{-2}$.

The observed temperature-dependence of the $^{103}$Rh relaxation rate constant $T_1^{-1}$ is shown for a field of $B \simeq 9.4\,\mathrm{T}$ in Figure 10(b). The rhodium $T_1^{-1}$ increases monotonically with increasing temperature over the relevant temperature range. At 315 K relaxation occurs with a time constant of $T_1(^{103}\mathrm{Rh}) = 30.6 \pm 1.1$ s.

A positive dependence of the $^{103}$Rh $T_1^{-1}$ on temperature was reported previously for $\mathrm{Rh(acac)_3}$ in solution (Benn et al.
160 (1985)).

## 6 Discussion

The temperature dependence of the $^{103}$Rh $T_1^{-1}$ as shown in Figure 10(b) indicates a dominant spin-rotation relaxation mechanism. For small molecules with a short rotational correlation time relative to the nuclear Larmor period, spin-rotation is the only mechanism that leads to a positive correlation of $T_1^{-1}$ with temperature (Hubbard (1963)). This is because the amplitudes of the
local magnetic fields generated by the spin-rotation interaction are proportional to the root-mean-square rotational angular mo-



mentum of the participating molecules – a quantity that is linked to the mean rotational kinetic energy of the molecules, which increases linearly with temperature. For other mechanisms, the decrease of the rotational correlation time $\tau_c$ with increasing temperature leads to a decrease in the relaxation rate with increasing temperature, in the fast-motion limit.

The field dependence of the $^{103}$Rh $T_1^{-1}$ as shown in Figure 10(a) displays a modest increase in relaxation rate with increasing magnetic field at high field, which suggests an additional contribution from the rotational modulation of the $^{103}$Rh chemical shift anisotropy (CSA) tensor. A finite $^{103}$Rh CSA tensor is symmetry-allowed under the $D_3$ point group of $Rh(acac)_3$ (Buckingham and Malm (1971)). Indeed, solid-state NMR data indicate a $^{103}$Rh shielding anisotropy of $\Delta\sigma \simeq -460\,\mathrm{ppm}$, with relativistic quantum chemistry calculations in reasonable agreement (Holmes et al. (2023)). The magnitude of this CSA tensor is modest by $^{103}$Rh standards. For example, the $^{103}$Rh nuclei in Rh paddlewheel complexes have a shielding anisotropy of 175 $|\Delta\sigma| \sim 9900\,\mathrm{ppm}$ (Harbor-Collins et al. (2023)).

In summary, we have demonstrated the successful transfer of polarisation between the central $^{103}$Rh nucleus and the three methine $^1$H nuclei in $Rh(acac)_3$, through the very small four-bond $^1$H-$^{103}$Rh couplings. The polarization transfer is more efficient for DualPol than for refocused-INEPT, even though the theoretical efficiency of DualPol is slightly less than that of refocused-INEPT, for the relevant $I_3S$ spin system. We have successfully exploited $^1$H-$^{103}$Rh polarization transfer to study 180 the longitudinal and transverse relaxation of $^{103}$Rh for $Rh(acac)_3$ in solution. The $^{103}$Rh $T_1$ relaxation is dominated by the spin-rotation mechanism, with a significant additional contribution from the $^{103}$Rh CSA at high magnetic field.

*Author contributions.* **Harry Harbor-Collins**: Conceptualization (equal); Data curation (equal); Formal analysis (equal); Investigation (equal); Methodology (equal); Software (equal); Validation (equal); Visualization (equal); Writing – original draft (equal); Writing – review & editing (equal). **Mohamed Sabba**: Conceptualization (equal); Data curation (equal); Formal analysis (equal); Investigation (equal); 185 Methodology (equal); Software (equal); Visualization (equal); Writing – original draft (equal); Writing – review & editing (equal). **Markus Leutzsch**: Conceptualization (equal); Funding acquisition (equal); Investigation (equal); Validation (equal); Writing review & editing (equal). **Malcolm H. Levitt**: Conceptualization (equal); Formal analysis (equal); Funding acquisition (equal); Investigation (equal); Project administration (equal); Resources (equal); Supervision (lead); Writing – original draft (equal); Writing review & editing (equal).

*Competing interests.* At least one of the (co-)authors is a member of the editorial board of Magnetic Resonance

*Acknowledgements.* We acknowledge funding received by the European Research Council (grant 786707-FunMagResBeacons), and EPSRC-UK (grants EP/P009980/1, EP/P030491/1, EP/V055593/1).





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
