# Peer review of "$^{1}H$ enhanced $^{103}Rh$ NMR spectroscopy and relaxometry of $Rh(acac)_{3}$ in solution"

_Magnetic Resonance, 2024_

## Author Response (AR1)

Thank you for giving us an opportunity to submit a revised manuscript to your Journal.
We greatly appreciate both the time and effort dedicated by yourself and the reviewers
in preparing feedback for our manuscript. Please see below for a point-to-point
response to the reviewers' comments.

Reviewer comment 1

I was not familiar with the previous work from this team on 103Rh NMR and the
polarisation transfer techniques used here, so I am looking at this "fresh". It seems to
be a sound piece of work, if rather specialised, and it is always pleasing to see graphs
that are properly drawn and labeled and numbers and units used with care.

One thing that particularly interested me, because it is an area I am currently looking at,
was the use of composite 180° pulses. How was the "symmetrized BB1" chosen? I've
just done a quick simulation and it appears that the symmetrized version yields a larger
phase dispersion during refocusing than the original BB1 pulse! However, in contrast,
an *antisymmetric* 180° pulse yields no phase dispersion whatsoever when used for
refocusing (JMR **214**, 68-75 (2012)) and that is the type of pulse I would have tried first -
for example, the F1 or G1 composite pulse (JMR **93**, 199-206 (1991)) or, if dual-
compensation was required, perhaps one of the ASBO family of pulses (JMR **225**, 81-92
(2012)). On the other hand, it is possible that in this particular application, the precise
choice of sequence is not that important, with the matching of the 1H and 103Rh field
strengths being the key issue?

**Author Response:** We thank the reviewer for their positive comments on our paper! In
this particular paper - chiefly a relaxometry study in which the dual-channel PulsePol or
DualPol sequence happened to be the cross-polarization sequence of choice - we have
not investigated the aspect of composite pulses in great depth, and this is a theme for
future research.

Nevertheless, sharing the reviewer's passion for composite pulses, and in the spirit of
transparency and open discussion in this journal, I will offer a longer answer below, in
candid detail, in a personal capacity.

In this work, composite 180 pulses are used in two places: both the cross-polarization
(DualPol) sequences, as well as the simple spin-echo train used to measure the $T_2$.
While both of these utilize a basic spin echo building block, the behavior of composite
pulses is not necessarily identical.

As the reviewer has correctly pointed out, symmetric composite pulses within spin
echo trains designed to refocus transverse magnetization are indeed associated with
phase distortions, a problem that was first examined by Levitt and Freeman
(https://doi.org/10.1016/0022-2364(81)90082-2) during the development of solution-
state decoupling sequences. As shown in their early work, the phase distortion cancels
out when an even number of spin echoes is used! Care was indeed taken to ensure all
experiments used an even number of spin echoes.

Coterminously with the aforementioned paper, Levitt and Freeman also proposed the
usage of supercycles (consisting of a pattern of Pi phase shifts) in spin echo trains,
combined with (symmetric) composite 180 pulses, as an additional layer of error
compensation (https://doi.org/10.1016/0022-2364(82)90042-7, see also Levitt's PhD
thesis https://ora.ox.ac.uk/objects/uuid:8365b030-de70-4b96-a9a0-
e0fd32290551/files/m854633531469ac51aff713a29e76ab78). This is important;
supercycles (MLEV-64) were used both in the $T_2$ sequence in our paper, and arguably
the DualPol sequence.

The reviewer also points out the usage of antisymmetric (which has become a synonym
with phase-distortionless) composite pulses, that is, composite pulses whose phase
shifts are antisymmetric in time. Composite pulses of the phase-distortionless variety
were first examined by the Pines group (https://doi.org/10.1016/0022-2364(85)90270-7)
with the theory explicitly laid out in the paper by Tycko, Pines, and Guckenheimer on
iterative schemes (see Appendix B of https://doi.org/10.1063/1.449228). Wimperis
significantly expanded previous work of the Pines group in the 1990s in a series of
papers, the most relevant of which described a class of composite pulses which
includes BB1 (https://doi.org/10.1006/jmra.1994.1159), as well as the closely related
F1 composite pulse (https://doi.org/10.1016/0022-2364(91)90043-S). As shown in the
work by Sami Husain, Minaru Kawamura, and Jonathan Jones
(https://doi.org/10.1016/j.jmr.2013.02.007), the BB1 and F1 composite pulses are
closely related, and there are subtle advantages to the usage of the symmetrized BB1
sequence (used in our paper) which include better off-resonance performance. More
recent work by Wimperis that the reviewer points out includes the description of the
ASBO-11 composite pulses (https://doi.org/10.1016/j.jmr.2012.10.003; see also the
PhD thesis of Smita Odedra https://theses.gla.ac.uk/5772/).

As for "why this composite pulse" – at the risk of sounding anecdotal – I investigated a
variety of composite pulses (including BB1, F1, ASBO-11, and many more) during my
PhD, working at the time with spin echo-based sequences that were severely sensitive
to pulse strength/rf homogeneity errors. In a nutshell, the symmetrized BB1 sequence
struck a perfect balance between efficiency improvements, elegance, and simplicity.
Note that the last two qualities are entirely subjective. We were also attracted by the
fact that the BB1 composite pulse could be used to replace 90 degree pulses as well as
54.74 degree pulses (used in e.g. the $T_{00}$ filters of singlet NMR) - we could use a single
family of composite pulses in any of our pulse sequences.

Directly after my PhD work on sequence development, we discovered that the PulsePol
sequence - invented by Benedikt Tratzmiller of the Ulm group (https://oparu.uni-
ulm.de/items/d5648138-630c-44a8-95e7-6b9fddde4a4a) for the purpose of optical
DNP in NV centres – was a rather effective sequence for the apparently unrelated
purpose of nuclear singlet excitation. It is worth noting that the Ulm group also
investigated the effect of composite pulses in the PulsePol sequence
(https://www.science.org/doi/10.1126/sciadv.aat8978). At this point, I decided to
compare the performance of a few composite pulses incorporated within PulsePol by
theory and experiment, including BB1, ASBO-11, and F1 (unpublished) whose
robustness comparisons may be found in our paper (https://pubs.aip.org/aip/jcp/article/157/13/134302/2841905). To my surprise, the use
of the (antisymmetric) F1 pulse did not appear to have any great difference in
performance to the (symmetrized) BB1 pulse in our particular implementations at the
time. As for the reasons - I am not prepared to speculate further.

It is worth noting that the PulsePol sequence has what can be argued to be a "built-in"
supercycle $[0,\pi,0,\pi]$ applied to the 180 pulses we have called "riffling"
(https://pubs.aip.org/aip/jcp/article/157/13/134302/2841905). Other (unpublished)
possibilities include $[0,\pi,0,\pi,\pi,0,\pi,0]$. These phase modifications were shown to be
compulsory for the final robustness of the sequence, regardless of whether symmetric
or antisymmetric composite pulses were used.

This success with the PulsePol sequence for robust singlet excitation inspired us to
adapt it as a heteronuclear cross-polarization sequence ("DualPol"), and I chose to use
the symmetrized BB1 pulse in light of the above history.

As for the reviewer's last point – it is worth noting that the matching of the $^1$H/$^{103}$Rh (I/S)
field strengths is **completely unnecessary** in a windowed cross-polarization sequence
such as DualPol! Unlike the traditional Hartmann-Hahn experiment (in which there is an
extraordinarily stringent condition on the matching of the strengths of the synchronous
rf fields), the I-S mixing in the DualPol sequence occurs entirely during the pulse-
interrupted free evolution. That is to say, the sequence would still work if the nutation
frequency on the I spins was 20,000 kHz and the nutation frequency on the S-spin
channel was 4 kHz, under the tacit assumption that both nutation frequencies greatly
exceed the IS J-coupling. We intentionally chose to match the $^1$H/$^{103}$Rh field strengths
largely for reasons of readability. The consequences of a matched vs. unmatched rf
field on the final performance of the sequence are unknown, and will be a theme for
future research.

No changes to the manuscript necessary.
Reviewer comment 2
This is a nice work describing field-dependent relaxation of Rh spins in the Rh(acac)3
complex. Rh signals were also detected indirectly using the DualPol sequence. The
main result is that a CSA tensor value was extracted and compared to published values
from computation, and spi-rotation was identified as a major relaxation mechanism as
well.

I do not have any comments, except a very positive one: this is very nice and complete
work and I fully support publication.

**Author Response:** We thank the reviewer for their very positive comments on our
manuscript! Just to clarify one point, in our work we do not estimate the Rh(acac)$_3$$^{103}$Rh
CSA value directly; instead, we observe that the $^{103}$Rh $T_1$ relaxation behaviour as a function of field strength is in general agreement with a prior measurement of the

Rh(acac)$_3$ $^{103}$Rh CSA (doi:10.1039/D3SC06026H).

No changes to the manuscript necessary.